# Effect of La Doping and Al Species on Bastnaesite Flotation: A Density Functional Theory Study

Xiancheng Shao [1], Guoyuan Wu [1], Gongliang Jiang [1], Ye Wang [1], Shikun Pu [2], Yaozhong Lan [1,*] and Dengbang Jiang [3,*]

[1] School of Materials and Energy, Yunnan University, Kunming 650091, China; sxcym036@163.com (X.S.); wgy66@tom.com (G.W.); jianggongliang6@163.com (G.J.); ignitionformula@163.com (Y.W.)
[2] Yunnan Lincang Xinyuan Germanium Industry Co., Ltd., Lincang 677000, China; 13988345130@163.com
[3] Green Preparation Technology of Biobased Materials National & Local Joint Engineering Research Center, Yunnan Minzu University, Kunming 650500, China
[*] Correspondence: yzhlan@ynu.edu.cn (Y.L.); 041814@ymu.edu.cn (D.J.)

**Abstract:** The recovery of rare earth elements from ores is crucial because of their applications in modern technology. Bastnaesite ($La/Ce(CO_3)F$) is typically found in deposits with other gangue minerals but can be purified by flotation. Accordingly, we investigated the interactions of the collector nonyl hydroxamic acid (NHA) with bastnaesite using density functional theory (DFT) calculations. In addition, we replaced Ce sites on the bastnaesite (100) surface with La and investigated the effect on NHA adsorption. Finally, we examined the effects of co-present aluminum species, which are frequently used inhibitors for associated gangue minerals during bastnaesite flotation, on NHA adsorption and, thus, the flotation efficiency. We found that doping with La increased the strength of adsorption between NHA and the bastnaesite (100) surface. In addition, we found that $Al(OH)_{3(s)}$ was adsorbed more strongly than NHA. Consequently, when $Al(OH)_{3(s)}$ is present in the flotation pulp, it is preferentially adsorbed, which reduces the number of sites for NHA adsorption and its flotation efficiency. These findings suggest that La doping can enhance the recovery of bastnaesite and indicate that the presence of Al minerals should be minimized.

**Keywords:** bastnaesite; aluminum ion; nonyl hydroxamic acid; density functional theory; atomic substitution

## 1. Introduction

Rare earth elements are widely used in aviation, military technologies, metallurgy, and other important fields because of their unique electronic structures as well as their intriguing optical, magnetic, and chemical properties [1,2]. Of the rare-earth-containing minerals, bastnaesite is the main source of Ce [3]. However, bastnaesite is found in several gangue minerals. Consequently, beneficiation must be carried out before the extraction of rare earth elements can be carried out [4]. Flotation is one of the most effective beneficiation methods for separating and enriching the target minerals [5–7]. During mineral processing, the minerals are crushed and ground, chemicals are added, and the minerals are dissolved; as a result, a complex mixture of metal ions is formed in the slurry [8,9]. Rare earth minerals are naturally hydrophilic, and the use of amphiphilic collectors can enhance the hydrophobicity of the particle surfaces, which is conducive to trapping ore particles on bubbles. Typical beneficiation collectors include hydroxamic acid, fatty acids, and organic phosphoric acids [10,11]. Based on the literature, hydroxamic acids are some of the most commonly used collectors for flotation during beneficiation and show excellent recovery rates [12]. Crucially, the two O atoms in hydroxamic acid easily form a stable five-membered chelate ring with rare earth metal cations [13].

During geological and mineralization processes, Ce is easily replaced by La, which has similar physicochemical properties [14–16]. Numerous studies have shown that the

substitution of mineral ions has a significant effect on flotation. For example, Zhu et al. reported [17] that Fe replaces Al on the spodumene (110) surface during grinding, subsequently becoming the key adsorption site for the collector and increasing the flotation recovery rate. Further, Chen et al. [18] found that Fe, Cd, Mn, and Cu easily replace Zn on the sphalerite surface and promote the oxidation of S, thereby improving the flotation recovery.

In addition to atomic substitution, the adsorption of metal ions present in the slurry on the surface of mineral ore particles has a significant effect on flotation. For example, Pan et al. reported [8] that the addition of lead ions to a slurry of ilmenite improved the flotation recovery. In contrast, Deng et al. reported [19] that the presence of iron ions inhibited the flotation of rhodochrosite.

The main metal ions in bastnaesite pulp are $Ca^{2+}$, $Mg^{2+}$, $Sr^{2+}$, $Al^{3+}$, and $Fe^{3+}$, and of these, only $Sr^{2+}$, $Al^{3+}$, and $Fe^{3+}$ are adsorbed on the bastnaesite surface [20]. However, $Sr^{2+}$ ions are only adsorbed on the bastnaesite surface in the form of Ce–O–Sr under strongly alkaline conditions. The effect of $Fe^{3+}$ on the flotation of bastnaesite has been studied in depth by Xinyu et al. [21]. In contrast, the effect of $Al^{3+}$ on the flotation of bastnaesite has not been studied thoroughly. Crucially, bastnaesite is often found in association with monazite, and aluminum salts are often introduced into the bastnaesite slurry as monazite inhibitors [22].

On the basis of previous research, atomic substitution and ion adsorption can affect the flotation of bastnaesite significantly. Therefore, a detailed study of their mechanisms of action could suggest ways to enhance the flotation of bastnaesite. Density functional theory (DFT) calculations are widely used as they have proven to be effective for studying the interactions between collectors and minerals, providing microscopic information on mineral–flotation agent interactions at the atomic and electronic levels. For example, Cao et al. [23] conducted DFT calculations to study the effects of the use of benzohydroxamic acid (BHA) and salicylic hydroxamic acid (SHA) as collectors on the flotation of bastnaesite, and Chen et al. [24] investigated the effects of the orbital symmetry matching of flotation agents with minerals.

Therefore, in this study, we performed DFT calculations to investigate the effect of La atom substitution and aluminum ion adsorption on the bastnaesite surface on its flotation, which to date, remains understudied. As a result, we reveal the effects of these factors on the flotation behavior from a microscopic perspective and provide suggestions for enhancing the recovery of bastnaesite.

## 2. Materials and Methods

### 2.1. Calculation Model

The crystal structure of bastnaesite was obtained from the Materials Project database, and the unit cell is shown in Figure 1a. The presence of NHA in the flotation caused a solution of NHA ions. Therefore, the NHA ion model was constructed in the simulation calculation of this paper (Figure 1b). Bastnaesite ($CeCO_3F$) crystallizes in the hexagonal space group *P*-62*c*, and the unit cell parameters are $a = b = 0.716$ nm, $c = 0.986$ nm, $\alpha = \beta = 90°$, and $\gamma = 120°$. The computational model includes three bastnaesite molecules. As shown, the crystal structure contains layers alternating between carbonate anions and Ce and F ions; the coordination number of Ce is nine, and the Ce ion coordinates with three F anions in the Ce—F layer and six O anions in the $CO_3F$ layer. In previous studies, surface energy was calculated for the different surfaces of bastnaesite so that the lowest energy (100) surface was used as the main cleavage surface. Figure 2 shows the bastnaesite (100) surface hydroxylation. According to the literature research, in alkaline solutions, the surface of bastnaesite is dihydroxylated [25]. Moreover, according to the principle of minimum energy, the dihydroxylation of the surface of bastnaesite is more stable than the monohydroxylation. Therefore, we constructed and optimized the dihydroxylated structure on the surface of bastnaesite as the surface to be adsorbed.

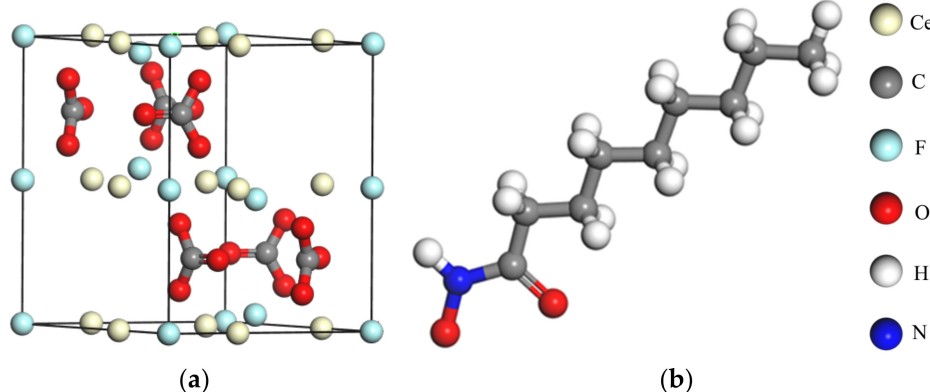

**Figure 1.** (**a**) Unit cell of bastnaesite and (**b**) the molecular model of NHA⁻.

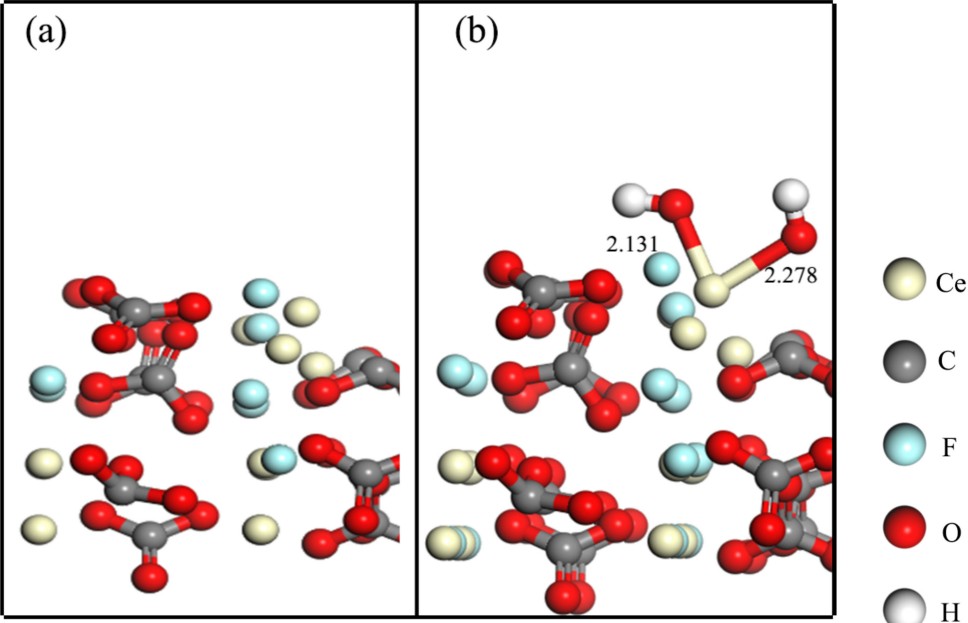

**Figure 2.** The (100) surface (**a**) and hydroxylated (100) surface (**b**) of bastnaesite.

### 2.2. Calculations

All calculations were performed in CASTEP and DMol3 in Materials Studio 2020 [26–29]. The first principle plane wave ultra soft pseudopotential (USPP) method based on the density functional theory (DFT) was used for the calculation. Generalized gradient approximation (GGA), ultrasoft pseudopotential and exchange correlation function Perdew-Burke-Ernzerhof for solid (PBEsol) were also used for simulation calculations [30,31]. PBEsol improved the calculation of bond length and lattice parameters on the basis of PBE. The initial cutoff energy was 380 eV, and the *k*-point grid was $3 \times 3 \times 2$. The energy convergence criterion was set to $1.0 \times 10^{-5}$ eV/atom, the maximum displacement tolerance was set to 0.001 Å, the maximum force tolerance was 0.05 eV/Å, and the self-consistent field convergence error was limited to $1.0 \times 10^{-6}$ eV/atom.

### 2.3. Calculation of Adsorption Energy

The magnitude of the adsorption energy can be used to measure the selectivity and stability of the adsorption of the agent on the mineral surface. Importantly, when the adsorption energy is negative, the interaction is favorable, and the lower the value, the more stable and selective the interaction between the flotation agent and mineral. However, when the adsorption energy is greater than or equal to zero, the adsorption is unstable or

cannot occur [32]. The adsorption energy ($\Delta E_{ads}$) was calculated using Equation (1) [33,34].

$$\Delta E_{ads} = E_{complex} - E_{collectors} - E_{mineral} \tag{1}$$

Here, $E_{complex}$ is the total energy of the complex after adsorption, and $E_{collectors}$ and $E_{mineral}$ are the energies of the collector molecules and mineral surface structures, respectively.

## 3. Results

### 3.1. Adsorption of NHA on the Bastnaesite (100) Surface

3.1.1. Adsorption Configuration of NHA on the Bastnaesite (100) Surface

As shown in Figure 3, NHA is adsorbed via the formation of a chelate structure with Ce on the (100) surface, as reported previously [35]. The calculated adsorption energy between NHA and the bastnaesite (100) surface was −536.370 kJ/mol. According to the calculation of the adsorption sites of the NHA bastnaesite (100) surface, it was found that the adsorption sites of the Ce atom on the boundary of the crystal plane are not as stable as the Ce atomic sites in the middle.

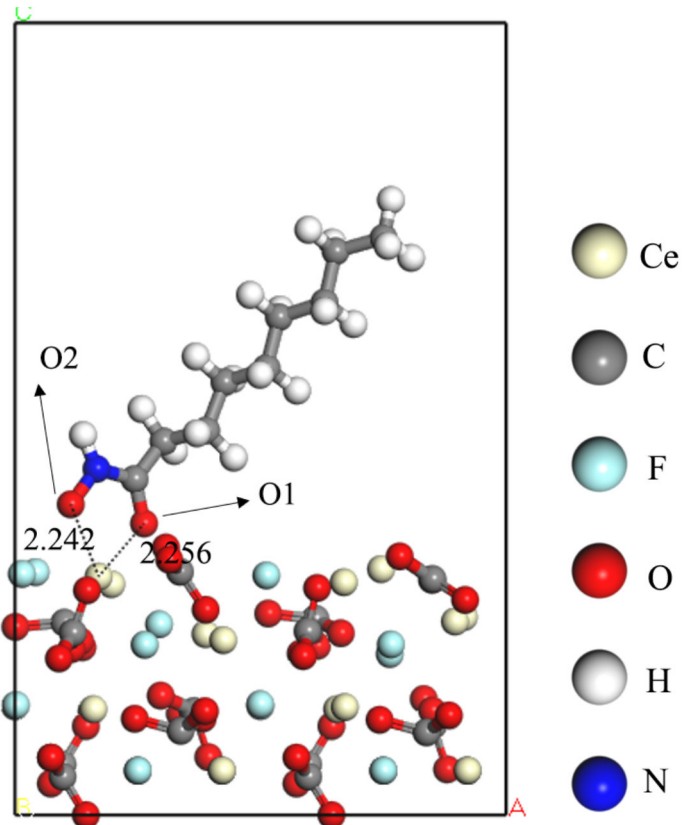

**Figure 3.** Coordination of NHA via the formation of a five-membered chelate ring with Ce on the (100) surface.

3.1.2. Electronic Structure and Population Analysis

The mechanism of the adsorption of NHA on the bastnaesite (100) surface was determined using Mulliken population analysis (Table 1) as well as the changes in the density of states before and after the adsorption of the oxygen atoms of NHA (labeled O1 for the oxime oxygen and O2 for the carbonyl oxygen) and Ce on the bastnaesite (100) surface (labeled Ce1).

**Table 1.** Mulliken populations of O and Ce before and after NHA adsorption on the bastnaesite (100) surface.

| Atoms | State | Mulliken Population/e | | | | | | Total Change (e) |
|---|---|---|---|---|---|---|---|---|
| | | s | p | d | f | Total | Charge | |
| O1 | Before adsorption | 1.84 | 4.74 | 0.00 | 0.00 | 6.59 | −0.59 | −0.01 |
| | After adsorption | 1.82 | 4.78 | 0.00 | 0.00 | 6.60 | −0.60 | |
| O2 | Before adsorption | 1.86 | 4.73 | 0.00 | 0.00 | 6.59 | −0.59 | −0.19 |
| | After adsorption | 1.85 | 4.92 | 0.00 | 0.00 | 6.77 | −0.77 | |
| Ce1 | Before adsorption | 2.19 | 6.02 | 1.30 | 1.15 | 10.65 | +1.35 | +0.24 |
| | After adsorption | 2.15 | 5.92 | 1.32 | 1.05 | 10.44 | +1.59 | |

Mulliken population analysis provides information concerning electron transfer as well as bonding interactions [36]. As shown in Table 1, the population of the 5d orbital of Ce increases, whereas those of the 6s, 4f, and 5p orbitals decrease. These results reflect the interactions between the O of NHA and Ce. In particular, the degree of electron transfer between O2 and Ce2 is greater than that between O1 and Ce1, probably because of O2 has lone pair electron. Further, because O is more electronegative than Ce, there is a shift in the electron density toward O; thus, Ce loses electron density, whereas O gains electron density.

As shown in Table 2, the Mulliken populations corresponding to O–Ce bonds are greater than 0.2, indicating that NHA forms a covalent chelate structure with Ce; note that a larger Mulliken population indicates a greater degree of covalency. Thus, the Ce–O2 bond has a more covalent nature than the Ce–O1 bond. In addition, the bond length of the former is shorter than the latter, suggesting a stronger bonding interaction.

**Table 2.** Mulliken populations of Ce–O bonds in the chelate structure.

| Bonds. | Population | Length (Å) | Electron Transfer ($\Delta q$) |
|---|---|---|---|
| Ce—O1 | 0.21 | 2.256 | 0.54 e |
| Ce—O2 | 0.25 | 2.242 | |

### 3.1.3. Density of States and Charge Density Analysis

Next, the partial density of states (PDOS) was calculated based on the O $2s^2 2p^4$ and Ce $4f^1 5d^1$ valence electron configurations. The PDOS for the five-membered chelate structure on the bastnaesite (100) surface is shown in Figure 4. As shown in Figure 4a,b, the main DOS peaks near the Fermi level are the O 2p and Ce 5d orbitals, indicating that the reactivity of the O 2p orbital before adsorption is high. Figure 4c,d show that after the adsorption of NHA, the conduction band (−10 to 0 eV) mainly comprises the O 2p and Ce 5d orbitals. In particular, the DOS of the O 2p orbital shows significant changes: the peak narrows and shifts toward a higher energy. In addition, the degree of delocalization increases. Between −10 and 0 eV, the DOS of the O 2p and Ce 5d orbitals shows changes characteristic of the formation of bonds. Combined with the calculated charges listed in Table 2 and the Mulliken populations, in the monodentate adsorption model, O2—Ce1 transfers more charge during bond formation than O1—Ce1. Notably, the higher the degree of electron transfer, the stronger is the interaction.

In addition, bond formation can be identified using a charge density plot, as shown in Figure 5a. In the figure, an increased charge density is denoted by a change in color from blue to red (Figure 5). As shown in the figure, there is a distinct increase in the charge density between O and Ce, but the electron distribution between O2 and Ce is significantly higher than that of O1—Ce1, suggesting the formation of a covalent bond between O2 and Ce1. Again, this may be a result of the lone-pair electrons on O2. This result is consistent with the results of the population analysis of the key atoms. In Figure 5b, it can be seen that the adsorbed electrons of NHA adsorbed on the bastnaesite (100) surface

are mainly distributed on the $CO_3$ group and collector, and there is also an obvious electron distribution between the O atom on NHA and the Ce atom on the surface of bastnaesite.

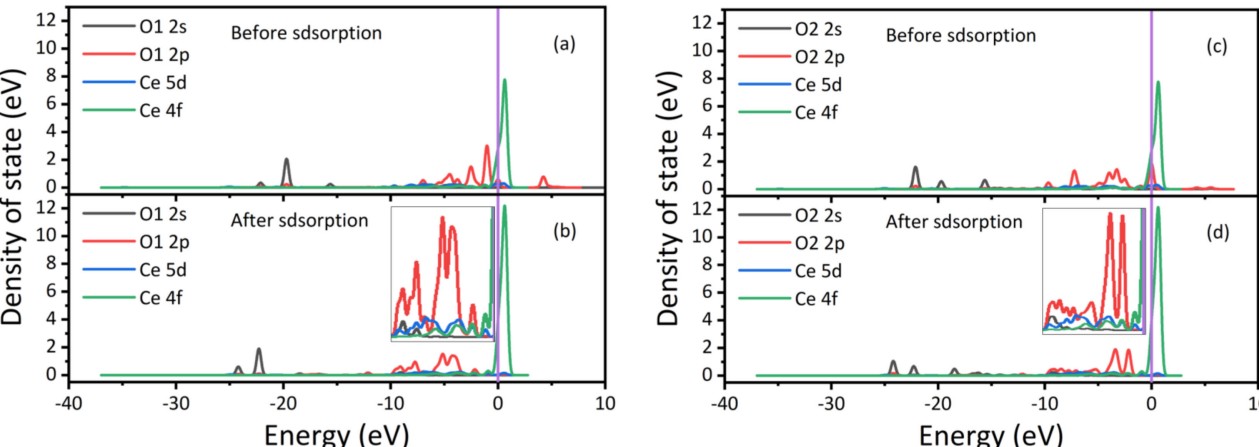

**Figure 4.** PDOS plots for O and Ce (**a**,**c**) before and (**b**,**d**) after the adsorption of NHA on the bastnaesite (100) surface.

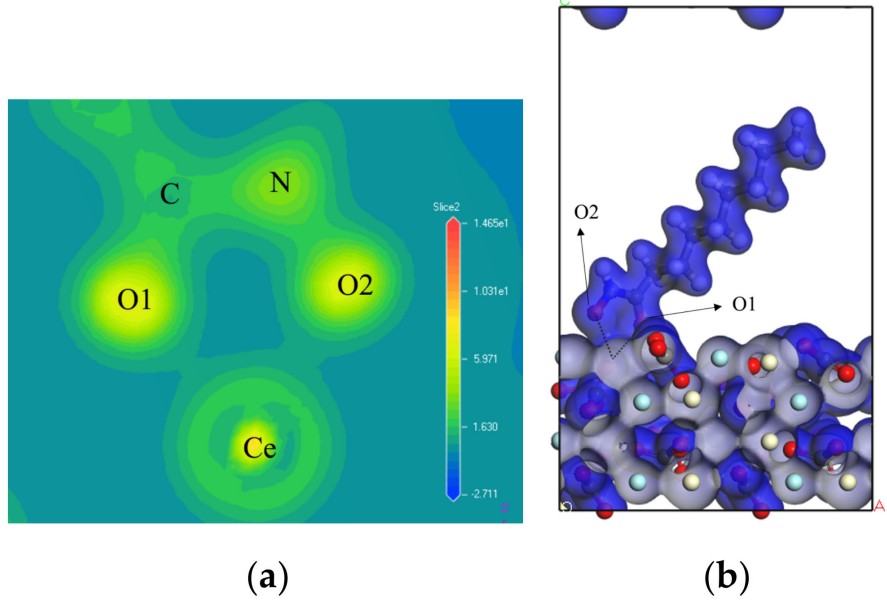

**(a)**                    **(b)**

**Figure 5.** (**a**) is the charge density of NHA adsorbed between O-Ce atoms on bastnaesite (100) surface, (**b**) is the charge equivalent surface of NHA adsorbed on the bastnaesite (100) surface.

### 3.2. Effect of the Substitution of Ce with La on the Adsorption of NHA on the Bastnaesite (100) Surface

#### 3.2.1. Adsorption of NHA on the La-Doped Bastnaesite (100) Surface

Atomic substitution usually occurs at the outermost layer of the cleavage surface, while it is more difficult at the inner layer. Figure 6 shows that there are four Ce atoms in the outermost layer of the bastnaesite (100) surface, but only two asymmetric Ce atoms. Studies on La doping mainly focus on these two asymmetric Ce atoms. The Mulliken populations of Ce1 after the introduction of La are listed in Table 3. As shown, the Ce atom has an increase in 5d electrons, and a decrease in 6s, 5p, and 4f electrons. The orbital electrons of Ce atom on the defect surface at position 1 and position 2 decrease by 0.05 e and 0.24 e, respectively.

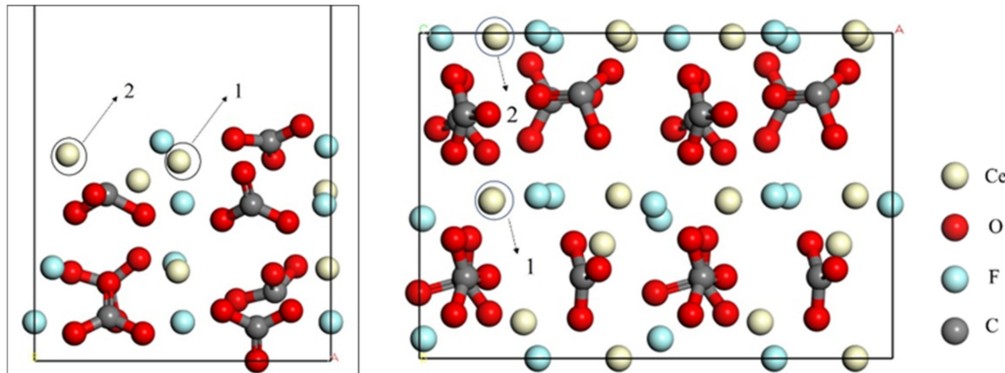

**Figure 6.** Two possible locations (1 and 2) for the substitution of Ce by La: (**left**) side view and (**right**) top view.

**Table 3.** The La atom replaces the Ce atom; the hydroxylated Mulliken population of the bastnaesite (100) surface.

| Atom | Substitution Position | State | s | p | d | f | Total (e) | Charge (e) | Total Shifts (e) |
|------|-----------------------|-------|------|------|------|------|-----------|------------|------------------|
| CE | 1 | Before | 2.19 | 6.02 | 1.30 | 1.14 | 10.65 | +1.35 | +0.05 |
| | | After | 2.15 | 5.97 | 1.35 | 1.13 | 10.60 | +1.40 | |
| | 2 | Before | 2.19 | 6.02 | 1.30 | 1.14 | 10.65 | +1.35 | +0.24 |
| | | After | 2.16 | 5.90 | 1.35 | 1.02 | 10.43 | +1.59 | |

Next, we investigated the effect of the substitution of Ce with La at positions 1 and 2 on the adsorption energies (see Figure 7 for the adsorption configurations). The adsorption energies were found to be −774.306 and −548.527 kJ/mol, respectively, representing reductions of 237.936 kJ/mol and 12.157 kJ/mol, respectively. In addition, the degrees of charge transfer and bonding increased after substitution at both sites (Table 4).

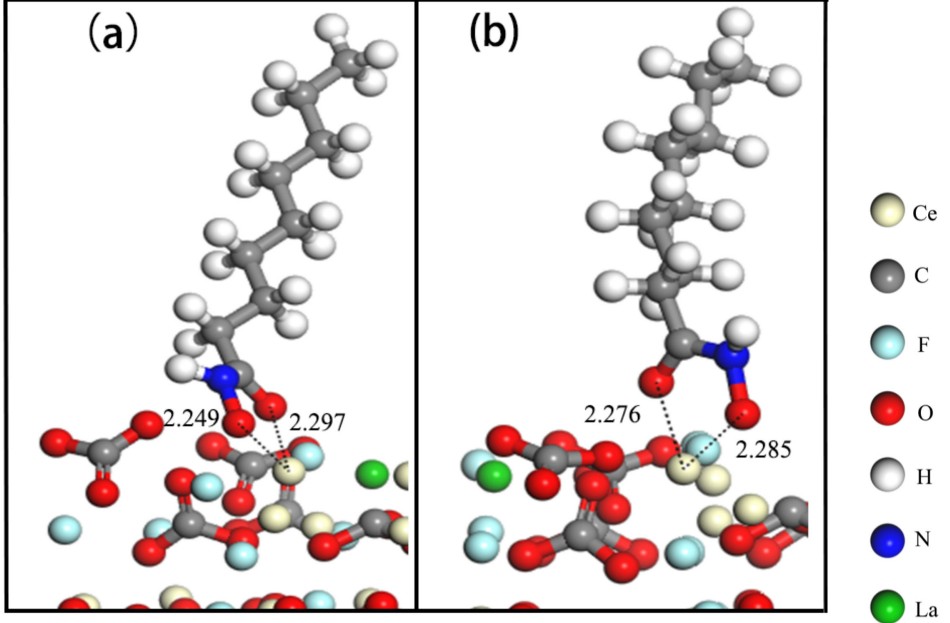

**Figure 7.** NHA adsorption on the doped bastnaesite (100) surface. La doping at (**a**) position 1 and (**b**) position 2.

**Table 4.** Effect of the replacement of Ce with La on the bastnaesite (100) surface on the adsorption energy.

| Substitution Position | Adsorption Energy/(kJ·mol$^{-1}$) | Electron Transfer ($\Delta q$) |
|---|---|---|
| 1 | $-774.306$ | 0.59e |
| 2 | $-548.527$ | 0.49$e$ |

By comparing the data in Tables 2 and 5, we found that the replacement of Ce by La increases the adsorption energy of NHA on the bastnaesite (100) surface. In addition, the O–Ce bond length increases and the Mulliken population of this bond decreases. Further, the electron density is biased toward O. Notably, the replacement of Ce by La at position 2 has a more significant effect on these values than that at position 1. Specifically, in the optimized structure after substitution at position 2, the O1—Ce (2.276 Å) bond length increases by 0.02 Å and the Mulliken population value is 0.22. In addition, the O2—Ce (2.285 Å) bond length increases by 0.043 Å and the Mulliken population (0.26) increases by 0.01. In contrast, after substitution at position 1, the Mulliken population of the O2—Ce bond is constant, but the population of O1—Ce increases by 0.02.

**Table 5.** Mulliken populations after the substitution of Ce by La at positions 1 and 2.

| Substitution Position | Bond | Population | Length (Å) |
|---|---|---|---|
| 1 | O1—Ce | 0.23 | 2.297 |
|   | O2—Ce | 0.25 | 2.249 |
| 2 | O1—Ce | 0.22 | 2.276 |
|   | O2—Ce | 0.26 | 2.285 |

### 3.2.2. Electronic Structure and Charge Density Analysis

Next, we used the Mulliken population analysis to investigate the surface charge transfer resulting from the addition of La to the bastnaesite (100) surface. By comparing the data in Tables 1, 3 and 6, the number of electrons transferred to O1 and O2 decreases. Thus, on substitution with La, the Ce adsorption sites lose electrons, and the strength of the interaction with O decreases.

**Table 6.** The Mulliken population analysis of NHA on the bastnaesite (100) surface after adsorption (monodentate).

| Substitution Position | State | Atom | s | p | d | f | Total (e) | Charge (e) | Total Charge (e) |
|---|---|---|---|---|---|---|---|---|---|
| 1 | | O1 | 1.81 | 4.79 | 0.00 | 0.00 | 6.60 | $-6.60$ | $-0.01$ |
| | | O2 | 1.81 | 4.85 | 0.00 | 0.00 | 6.66 | $-0.66$ | $-0.07$ |
| | Before adsorption | Ce | 2.15 | 5.97 | 1.35 | 1.13 | 10.65 | $+1.35$ | $+0.12$ |
| | After adsorption | Ce | 2.14 | 5.90 | 1.40 | 1.09 | 10.53 | $+1.47$ | |
| 2 | | O1 | 1.81 | 4.78 | 0.00 | 0.00 | 6.59 | $-0.59$ | $+0.00$ |
| | | O2 | 1.85 | 4.91 | 0.00 | 0.00 | 6.76 | $-0.76$ | $-0.17$ |
| | Before adsorption | Ce | 2.16 | 5.90 | 1.35 | 1.02 | 10.43 | 1.57 | $+0.01$ |
| | After adsorption | Ce | 2.14 | 5.88 | 1.38 | 1.02 | 10.42 | $+1.58$ | |

According to Figure 8, when NHA adsorbates on the surface of defective bastnaesite, electrons in the O 2p orbital move towards high energy, which is consistent with the previous population analysis.

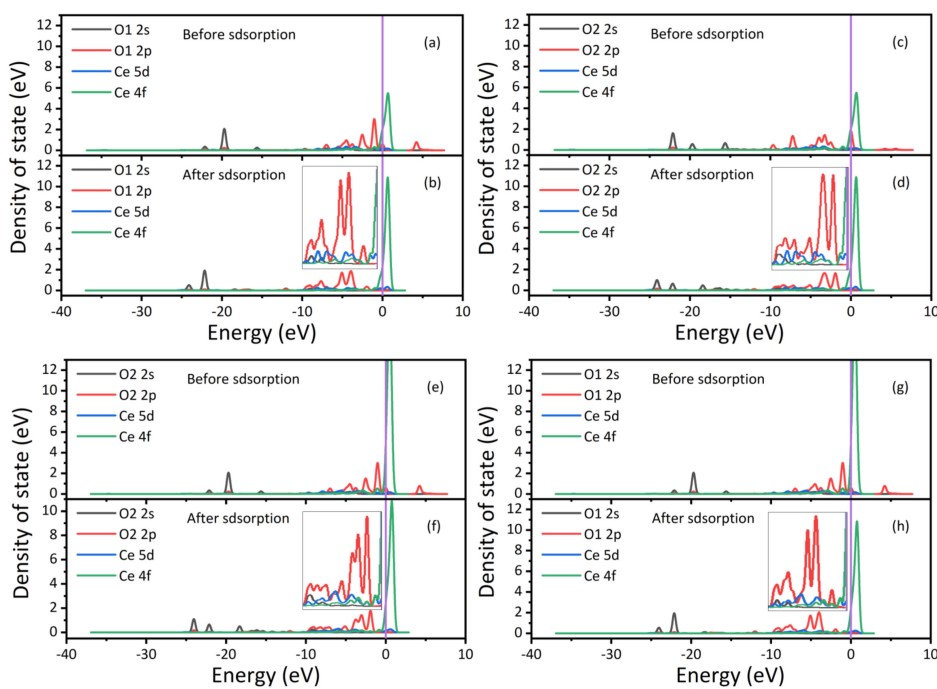

**Figure 8.** Density-of-state plots after the adsorption of NHA on the La-doped bastnaesite (100) surface: (**a**–**d**) substitution at position 1; (**e**–**h**) substitution at position 2.

As shown in Figure 9c,d, except for O2–Ce, when position 1 is substituted by La, the electron density in the Ce–O bond is mainly biased toward O atoms, resulting in no apparent electron overlap between the bonds, which is consistent with the bond layout analysis results.

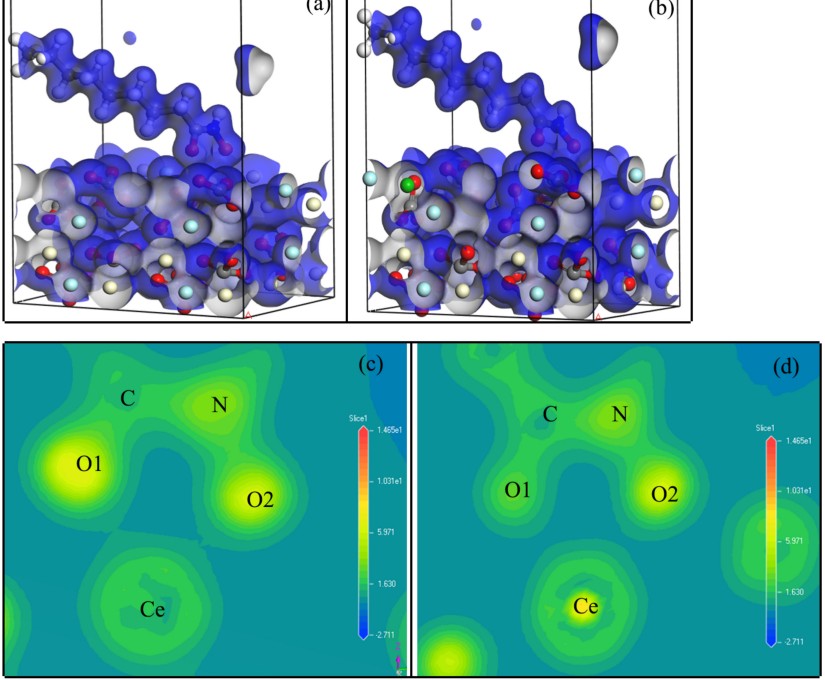

**Figure 9.** (**a**,**b**) are the charge density isoplanes adsorbed by NHA on the defective bastnaesite (100) surface; (**c**,**d**) are the charge densities between O—Ce atoms. (**a**,**c**) are the La atoms replacing Ce atoms at position 1, (**b**,**d**) are the La atoms replacing Ce atoms at position 2.

### 3.3. Effect of $Al^{3+}$ on NHA Adsorption

3.3.1. Adsorption of Aluminum Ions on the Bastnaesite (100) Surface

The recovery rate of bastnaesite flotation is the best when the pH is about 9 [37]. According to the solution chemistry, $Al(OH)_{3(s)}$ is the most stable species of aluminum ions in an alkaline solution, the Ce atoms on the surface of bastnaesite are also dehydroxylated $(Ce(OH)_2^+)$. $Al(OH)_{3(s)}$ can adsorb on a bastnaesite surface by complexing with $Ce(OH)_2^+$ in the form of Ce-O-Al-O-Ce via a dehydration reaction [38,39]. On the adsorption of $Al(OH)_3$ on the bastnaesite (100) surface, a four-membered ring chelate structure is formed, and the associated adsorption energy is calculated to be $-1188.910$ kJ/mol, twice that for NHA adsorption at Ce. Thus, $Al(OH)_{3(s)}$ is more easily adsorbed on the surface of bastnaesite than on NHA.

As shown in the adsorption model in Figure 10, both $Al(OH)_{3(s)}$ and NHA form chelate structures through oxygen atoms [20]. In Figure 10b, an adsorption model involving $Al(OH)_3$ and NHA is shown. Here, $Al(OH)_{3(s)}$ interacts with the Ce atom on the (100) surface, and NHA interacts with Al. As shown, the bond lengths between Al and the oxygen atoms in NHA are 1.831 and 1.804 Å, respectively. The adsorption energy of NHA⁻ on the surface of bastnaesite coated with $Al(OH)_{3(s)}$ is $-604.972$ kJ/mol, significantly higher than that of NHA on pristine bastnaesite. Because the Al atom radius is smaller than the Ce atom, the NHA adsorbed Al atom bond is more stable. Thus, NHA was preferentially adsorbed on the Al atoms. Accordingly, in the presence of $Al(OH)_{3(s)}$, the adsorption of NHA on Ce sites is inhibited.

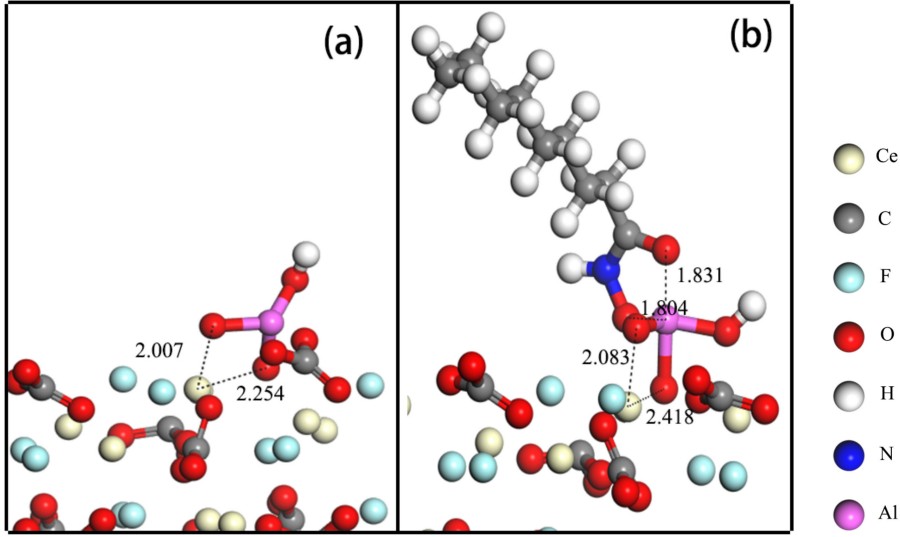

**Figure 10.** Adsorption of (**a**) $Al(OH)_{3(s)}$ and (**b**) $Al(OH)_3$ and NHA on the bastnaesite (100) surface.

3.3.2. Electronic Structure and Population Analysis

Next, we carried out the Mulliken population analysis of the $Al(OH)_{3(s)}$ adsorbed structure using the monodentate adsorption model; the results are listed in Table 7. The charge transfer to adsorbed $Al(OH)_{3(s)}$ is greater than that to NHA. As shown in the density of states plots in Figure 11, when $Al(OH)_{3(s)}$ is adsorbed, the O 2p orbital moves to higher energies. In addition, the Ce 4f peak increases in intensity and becomes sharp near the Fermi level, indicating a loss of electrons.

As shown in Table 8, the bonding degree between Ce and the O of $Al(OH)_{3(s)}$ is greater than 0.2, indicating covalency. Further, as shown in Figure 11b,d, the O 2p and Ce 5d orbitals overlap, also indicating the formation of a covalent bond between these elements.

**Table 7.** The Mulliken population analysis of the bastnaesite (100) surface after adsorption (bidentate).

| Atom | States | s | p | d | f | Total (e) | Charge (e) | Total Shifts (e) |
|------|--------|---|---|---|---|-----------|------------|------------------|
| O1 | Before adsorption | 1.97 | 4.74 | 0.00 | 0.00 | 6.71 | −0.70 | −0.13 |
|    | After adsorption | 1.89 | 4.94 | 0.00 | 0.00 | 6.83 | −0.83 | |
| O2 | Before adsorption | 1.97 | 4.74 | 0.00 | 0.00 | 6.71 | −0.70 | −0.18 |
|    | After adsorption | 1.86 | 5.02 | 0.00 | 0.00 | 6.88 | −0.88 | |
| Ce1 | Before adsorption | 2.19 | 6.02 | 1.30 | 1.14 | 10.65 | 1.35 | +0.19 |
|     | After adsorption | 2.13 | 5.91 | 1.38 | 1.04 | 10.46 | +1.54 | |

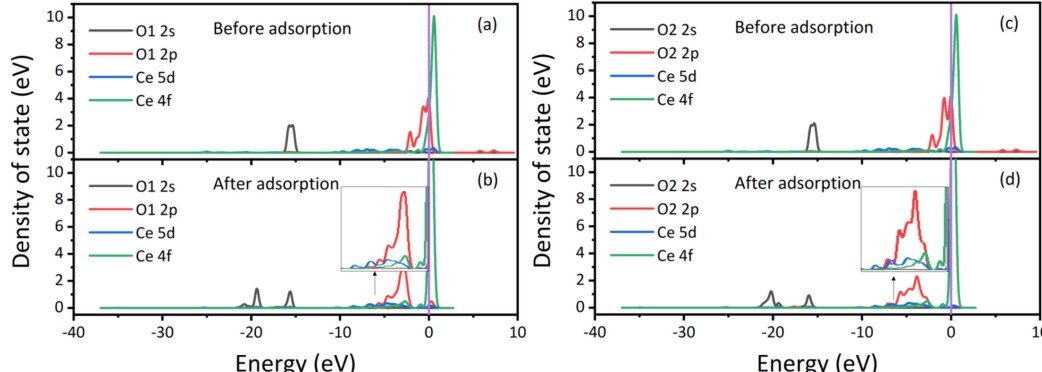

**Figure 11.** Density of states of Ce and O (**a**,**c**) before and (**b**,**d**) after the adsorption of $Al(OH)_{3(s)}$ on the bastnaesite (100) surface.

**Table 8.** Bond and population analyses for adsorption on the bastnaesite (100) surface (bidentate).

| Adsorbate | Bond | Population | Bond Length (Å) |
|-----------|------|-----------|-----------------|
| $Al(OH)_3$ | O1—Ce1 | 0.34 | 2.077 |
|            | O2—Ce1 | 0.20 | 2.254 |

In summary, when bastnaesite is treated with Al ions, the adsorption of NHA by Ce is inhibited and, instead, the adsorption of NHA occurs via adsorbed $Al(OH)_{3(s)}$. For example, in bidentate adsorption, at least one Ce site is occupied by an aluminum species [39]. Overall, compared to NHA, $Al(OH)_{3(s)}$ is preferentially adsorbed on the bastnaesite (100) surface.

## 4. Conclusions

In this paper, the implicit solvation model is used to study the reaction between the collector and bastnaesite itself. Of course, in the dressing itself, there are many explicit solvents that can affect adsorption. There must be some differences between the theoretical calculation and experiment of mineral processing. In this study, the adsorption of NHA on the bastnaesite (100) surface was investigated using DFT calculations. In addition, the effect of the substitution of Ce ions on the surface with La ions was investigated. Finally, we investigated the effect of the presence of $Al(OH)_{3(s)}$ on NHA adsorption. We found that doping with La increased the strength of interaction of NHA with the surface, suggesting a potential method for enhancing the flotation of bastnaesite. Further, in the presence of $Al(OH)_{3(s)}$, this species was preferentially adsorbed on the Ce sites via a four-membered chelate structure, thus reducing the number of sites for interaction with NHA and, consequently, inhibiting the flotation of bastnaesite.

**Author Contributions:** Conceptualization, D.J., G.W. and Y.L.; Data curation, X.S. and G.J.; Formal analysis, Y.W. and X.S.; Funding acquisition, D.J. and Y.L.; Investigation, D.J., G.W. and S.P.; Methodology, D.J., Y.W., G.J., X.S. and S.P.; Project administration, D.J. and Y.L.; Resources, S.P. and Y.L.; Validation, X.S.; Visualization, X.S.; Writing—original draft, X.S.; Writing—review and editing, D.J. All authors have read and agreed to the published version of the manuscript.

**Funding:** This research was funded by National Natural Science Foundation of China [grant numbers: U2002215] and Yunnan Provincial Science and Technology Department (grant numbers: 202001AU070007).

**Institutional Review Board Statement:** Not applicable.

**Informed Consent Statement:** Not applicable.

**Data Availability Statement:** The data presented in this study are available on request from the corresponding author.

**Acknowledgments:** This work was supported by National Natural Science Foundation of China and Yunnan Provincial Science and Technology Department. The authors would like to thank Ren Shudi from Shiyanjia Lab (www.shiyanjia.com) for the mechanical property analysis.

**Conflicts of Interest:** The authors declare no conflict of interest.

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
