# Peer review of "Effect of La Doping and Al Species on Bastnaesite Flotation: A Density Functional Theory Study"

_minerals, doi:10.3390/min13040583_

Round 1
Reviewer 1 Report
The manuscript focused on studying the effect of La doping and Al species on bastnaesite flotation by the density functional theory (DFT) method. The adsorption structure, Mulliken populations of atoms and bonds, PDOS plots, and electron density map are discussed in this paper. However, there are many problems with the content of the article, which should be modified.
The main comments:
1. This paper did not design a novel collector, and I think “ Hence, there is a need to develop new collectors and enhance the efficiency of existing collectors” should not appear in the Abstract Part.
2. In Fig.1(b), NHA is not the molecular structure but is the ionic form. It should be clearly stated in the content.
3. In line 105, Part 2.3, please explain why you choose the (100) surface. Did you calculate the surface energies or reference others?
4. Please mark clearly in the figures which atoms are O1 and O2 mentioned in Table 1.
5. In Fig.7, the colors of La and F are too similar to distinguish atoms. Plus, please discuss the difference between Position 1 and Position 2, and how did you determine the La doping positions?
6. In Fig.10, why did you put the AlO3H instead of Al(OH)3? Please give your reasons.
7. Al(OH)3-treated bastnaesite was calculated to be -604.972 kJ/mol
8. Please use the unified isosurface level in electron density maps and the imagers need higher resolution.
9. Improve all figures' resolution in this paper and adjust the font size to make pictures clearer.
10. Many formatting errors in the manuscript. For example, “Materials Studio 2020” in line 94, “O2p and Ce 5d” in Part 3.1.3, and Ref.5 in line 300, etc.
11. Recheck for grammatical errors.

Author Response
April 14, 2023
Dear Sir/Madam,
Thank you for reviewing our manuscript and offering valuable advice. In accordance with your suggestions, we have made the following revisions to our manuscript:
Point 1: This paper did not design a novel collector, and I think “Hence, there is a need to develop new collectors and enhance the efficiency of existing collectors” should not appear in the Abstract Part.
Response 1: Thank you for your suggestions. This sentence has been removed from the abstract of this article.
Point 2: In Fig.1(b), NHA is not the molecular structure but is the ionic form. It should be clearly stated in the content.
Response 2: Thank you for your suggestions. NHA exists in the flotation solution as NHA ions. Based on your suggestion, we have provided explanations in the manuscript.
Point 3: In line 105, Part 2.3, please explain why you choose the (100) surface. Did you calculate the surface energies or reference others?
Response 3: Thank you for your suggestions. The surface energy of different bastnaesite surfaces has been calculated by predecessors, and the results show that the surface energy of (100) is the lowest, making it the main cleavage surface of bastnaesite. We have added relevant references to the paper.
Point 4: Please mark clearly in the figures which atoms are O1 and O2 mentioned in Table 1
Response 4: Thank you for your suggestions. We have labeled the O1 and O2 atoms in Figure 3.
Point 5: In Fig.7, the colors of La and F are too similar to distinguish atoms. Plus, please discuss the difference between Position 1 and Position 2, and how did you determine the La doping positions?
Response 5: Thank you for your suggestions. The La atomic color has been modified in Figure 7. Atomic substitution usually occurs at the outermost layer of the cleavage surface, while it is more difficult at the inner layer. Figure 6 shows that there are four Ce atoms in the outermost layer of the bastnaesite (100) surface, but only two asymmetric Ce atoms. Studies on La doping mainly focus on these two asymmetric Ce atoms. We also explained this in our paper.
Point 6: In Fig.10, why did you put the AlO3H instead of Al(OH)3? Please give your reasons.
Response 6: Thank you for your suggestions. The recovery rate of bastnaesite flotation is the best when pH is about 9 2. According to the solution chemistry, Al(OH)3(s) is the most stable species of aluminum ions in alkaline solution, the Ce atoms on the surface of bastnaesite are also dehydroxylated (). Al(OH)3(s) can adsorb on bastnaesite surface by complexing with in the form of Ce-O-Al-O-Ce via dehydration reaction3, 4. We also explained this in our paper.
Point 7: Al(OH)3treated bastnaesite was calculated to be -604.972 kJ/mol
Response 7: Thank you for pointing out this error. We have changed it to” The adsorption energy of NHA- on the surface of bastnaesite coated with Al(OH)3 is -604.972 kJ/mol” in the paper.
Point 8: Please use the unified isosurface level in electron density maps and the imagers need higher resolution.
Response 8: Thank you for your suggestions. In this paper, the isoplanes of charge density graph have been unified and the resolution of the graph has been improved.
Point 9: Improve all figures' resolution in this paper and adjust the font size to make pictures clearer.
Response 9: Thank you for your suggestions. In this article, all the diagrams have been redrawn and used a uniform font.
Point 10: Many formatting errors in the manuscript. For example, “Materials Studio 2020” in line 94, “O2p and Ce 5d” in Part 3.1.3, and Ref.5 in line 300, etc
Response 10: Thank you for pointing out these errors.. We have corrected these errors.
Point 11: Recheck for grammatical errors.
Response 11: Thank you for pointing out these errors.. We further checked the manuscript and corrected some syntax error.
References:
- Pradip; Rai, B., Molecular modeling and rational design of flotation reagents. International Journal Of Mineral Processing 2003,72(1-4), 95-110.
- Jordens, A.; Marion, C.; Grammatikopoulos, T.; Hart, B.; Waters, K. E., Beneficiation of the Nechalacho rare earth deposit: Flotation response using benzohydroxamic acid. Minerals Engineering 2016,99, 158-169.
- Cao, S. M.; Cao, Y. J.; Ma, Z. L.; Liao, Y. F., The adsorption mechanism of Al(III) and Fe(III) ions on bastnaesite surfaces. Physicochemical Problems of Mineral Processing 2019,55(1), 97-107.
- Eskanlou, A.; Huang, Q.; Foucaud, Y.; Badawi, M.; Romero, A. H., Effect of Al3+ and Mg2+ on the flotation of fluorapatite using fatty- and hydroxamic-acid collectors – A multiscale investigation. Applied Surface Science 2022,572, 151499.
Reviewer 2 Report
In this manuscript, the authors use density functional theory to investigate the interaction of nonylhydroxamic acid (NHA) with bastnaesite, which is an important step in the recovery of rare earth elements. The effects of La doping and co-present Al-containing species were also investigated. Overall, I found the manuscript to be clearly written and easy to understand. However, as a non-expert in the field of minerals, I am unable to assess the significance of this work on the field and can only comment on the DFT calculations and models. In this regard, some aspects need clarification and potentially additional calculations to be performed:
- In the introduction, the abbreviations BHA and SHA should be defined for the benefit of readers.
- In Figure 1a, the labels Ce, O and F are not clearly distinguishable from the atoms and should be repositioned.
- The authors should explain why they used the (100) plane of bastnaesite for their simulations. I am also unsure if the pristine (100) surface is a realistic model. After the ore is broken down and floated in solution, there should be defect sites and hydroxyl groups on the surface, created by the interaction with water. The authors should comment further.
- Appropriate references for the software and functional used should be cited. The pseudopotentials used should also be stated.
- The adsorption energy was defined as the difference between the electronic energies of the product and the reactants. Why was zero point energy and entropy not considered? Also, the authors should comment if the lack of implicit or explicit solvent would greatly affect the results.
- In Figure 3, did the authors explore different adsorption sites to determine which is the most favorable?
- The structural model of Al(OH)3 adsorbed on bastnaesite in Figure 10 should be explained in more detail. The speciation of Al(OH)3 in aqueous solution is quite complex. How does the proposed Ce-O-Al-O-Ce four-membered ring form? It seems to require a double deprotonation of Al(OH)3, which would be very difficult under the conditions proposed.
- Finally, the initials of the last author in many of the references are incorrect. The authors should check the settings of their reference manager.
Author Response
April 14, 2023
Dear Sir/Madam,
Thank you for reviewing our manuscript and offering valuable advice. In accordance with your suggestions, we have made the following revisions to our manuscript:
Point 1:In the introduction, the abbreviations BHA and SHA should be defined for the benefit of readers.
Response 1: Thank you for the error you pointed out. We have amended benzohydroxamic acid(BHA) and salicylic hydroxamic acid(SHA) in the foreword.
Point 2: In Figure 1a, the labels Ce, O and F are not clearly distinguishable from the atoms and should be repositioned.
Response 2: Thank you for the error you pointed out. The fluocerite crystal model has been modified in Figure 1.
Point 3: The authors should explain why they used the (100) plane of bastnaesite for their simulations. I am also unsure if the pristine (100) surface is a realistic model. After the ore is broken down and floated in solution, there should be defect sites and hydroxyl groups on the surface, created by the interaction with water. The authors should comment further.
Response 3: Thank you for your valuable suggestion. We apologize that we did not clearly explain these details in our previous manuscript.
(1)The surface energy of different bastnaesite surfaces has been calculated by predecessors, and the results show that the surface energy of (100) is the lowest, making it the main cleavage surface of bastnaesite. We have added relevant references to the paper.
(2) Based on your suggestion, we have constructed a surface hydroxylation model for bastnaesite and conducted further calculations. The results show that, and the results show that Ce atoms on the surface of bastnaesite are also dehydroxylated Ce(OH)2+.
(3) This new model is closer to the real situation. As for defects, this manuscript only considers the replacement of La atoms. Due to the limited length of the article, no further research was conducted.
Point 4: Appropriate references for the software and functional used should be cited. The pseudopotentials used should also be stated.
Response 4: Thank you for your suggestions. The related literature of CASTEP and DMol3 in Materials Studio 2020 has been added to this paper2-4. Generalized gradient approximation (GGA) and exchange correlation function Perdew-Burke-Ernzerhof for solid (PBEsol) were used for simulation calculation and related data were added5, 6. The pseudopotential used in the calculation is Ultrasoft.
Point 5: The adsorption energy was defined as the difference between the electronic energies of the product and the reactants. Why was zero point energy and entropy not considered? Also, the authors should comment if the lack of implicit or explicit solvent would greatly affect the results.
Response 5: These are two very good questions.
(1)We use the final energy (Kohn Sham energy) calculated by castep as the total energy of each system before and after adsorption to calculate the adsorption energy. The final energy calculated by castep has taken into account zero energy correction. In our calculations, we did not consider entropy or calculate the adsorption free energy.
(2) The solvation effect has a significant impact on the adsorption energy of the system, but introducing solvent effect can lead to the calculation system being too large and difficult to converge. Therefore, due to the limitation of computational resources, our calculation did not consider the solvation effect. However, based on a large number of previous research results, it has been found that the DFT calculation results are still in good agreement with actual adsorption experiments without considering the solvation effect.
Point 6: In Figure 3, did the authors explore different adsorption sites to determine which is the most favorable?
Response 6: This is a very good question. Sorry, we did not provide a clear explanation of this exploration process in the manuscript. The outermost layer of the bastnaesite (100) surface has only two asymmetric Ce atoms. We calculated the adsorption energies of NHA on these two Ce atoms through optimization, and determined the most likely adsorption sites by comparing the values of adsorption energies.
Point 7: The structural model of Al(OH)3 adsorbed on bastnaesite in Figure 10 should be explained in more detail. The speciation of Al(OH)3 in aqueous solution is quite complex. How does the proposed Ce-O-Al-O-Ce four-membered ring form? It seems to require a double deprotonation of Al(OH)3, which would be very difficult under the conditions proposed.
Response 7: Thank you for the error you pointed out. Hydroxamic acid flotation bastnaesite has the best recovery at pH around 9 7. According to the solution chemistry, Al(OH)3(s) is the most stable species of aluminum ions in solution when pH=9, and can adsorb on bastnaesite surface by complexing with in the form of Ce-O-Al-O-Ce via dehydration reaction8, 9. In this paper, Al(OH)3(s) was calculated on the surface of bastnaesite after surface hydroxylation.
Point 8: Finally, the initials of the last author in many of the references are incorrect. The authors should check the settings of their reference manager.
Response 8: Thank you for pointing out this error. We have re edited the reference section based on your suggestion.
References:
- Pradip; Rai, B., Molecular modeling and rational design of flotation reagents. International Journal Of Mineral Processing 2003,72(1-4), 95-110.
- Delley, B., From molecules to solids with the DMol3 approach. The Journal of Chemical Physics 2000,113(18), 7756-7764.
- Segall, M. D.; Lindan, P. J. D.; Probert, M. J.; Pickard, C. J.; Hasnip, P. J.; Clark, S. J.; Payne, M. C., First-principles simulation: ideas, illustrations and the CASTEP code. Journal of Physics: Condensed Matter 2002,14(11), 2717-2744.
- Clark, S. J.; Segall, M. D.; Pickard, C. J.; Hasnip, P. J.; Probert, M. J.; Refson, K.; Payne, M. C., First principles methods using CASTEP. Zeitschrift für Kristallographie 2005,220(5-6), 567-570.
- Perdew, J. P.; Burke, K.; Ernzerhof, M., Comment on "Generalized gradient approximation made simple" - Reply. Physical Review Letters 1998,80(4), 891-891.
- Perdew, J. P. In Restoring the Density-Gradient Expansion for Exchange in a GGA for Solid and Surfaces, 2008 APS March Meeting, 2008.
- Jordens, A.; Marion, C.; Grammatikopoulos, T.; Hart, B.; Waters, K. E., Beneficiation of the Nechalacho rare earth deposit: Flotation response using benzohydroxamic acid. Minerals Engineering 2016,99, 158-169.
- Cao, S. M.; Cao, Y. J.; Ma, Z. L.; Liao, Y. F., The adsorption mechanism of Al(III) and Fe(III) ions on bastnaesite surfaces. Physicochemical Problems of Mineral Processing 2019,55(1), 97-107.
- Eskanlou, A.; Huang, Q.; Foucaud, Y.; Badawi, M.; Romero, A. H., Effect of Al3+ and Mg2+ on the flotation of fluorapatite using fatty- and hydroxamic-acid collectors – A multiscale investigation. Applied Surface Science 2022,572, 151499.
Round 2
Reviewer 2 Report
The authors have revised the ms according to the suggestions, and I can recommend publication.